# Electromagnetic Fields Exposure Assessment in Europe Utilizing Publicly Available Data

**DOI:** 10.3390/s22218481

**Published:** 2022-11-04

**Authors:** Serafeim Iakovidis, Christos Apostolidis, Athanasios Manassas, Theodoros Samaras

**Affiliations:** 1CIRI—Center for Interdisciplinary Research and Innovation, Aristotle University of Thessaloniki, 57001 Thermi, Greece; 2Radiocommunications Lab, Department of Physics, Aristotle University of Thessaloniki, 54124 Thessaloniki, Greece

**Keywords:** radiofrequency electromagnetic fields (RF-EMF), continuous RF-EMF monitoring networks, temporal exposure assessment, microenvironments, in situ measurements

## Abstract

The ever-increasing use of wireless communication systems during the last few decades has raised concerns about the potential health effects of electromagnetic fields (EMFs) on humans. Safety limits and exposure assessment methods were developed and are regularly updated to mitigate health risks. Continuous radiofrequency EMF monitoring networks and in situ measurement campaigns provide useful information about environmental EMF levels and their variations over time and in different microenvironments. In this study, published data from the five largest monitoring networks and from two extensive in situ measurement campaigns in different European countries were gathered and processed. Median electric field values for monitoring networks across different countries lay in the interval of 0.67–1.51 V/m. The median electric field value across different microenvironments, as evaluated from in situ measurements, varied from 0.10 V/m to 1.42 V/m. The differences between networks were identified and mainly attributed to variations in population density. No significant trends in the temporal evolution of EMF levels were observed. The influences of parameters such as population density, type of microenvironment, and height of measurement on EMF levels were investigated.

## 1. Introduction

The use of wireless communications has grown significantly during the last few decades, affecting our everyday lives in an unprecedented way. Along with this growth, there has been a rise in concerns about the potential health effects of electromagnetic fields (EMFs) on humans. In this context, guidelines from international organizations have been published [1] that recommend specific safety limits on EMF levels. Exposure limits at a national level vary among different countries or even within the same country (e.g., Belgium). Member states of the European Union (EU) may adopt the exposure limits of the Council Recommendation 1999/519/EC or stricter ones [2]. Many national and regional authorities in the EU have indeed introduced stricter limits, such as Greece [3] and the region of Brussels in Belgium [4]. EMFs’ interaction with the human body differs across the frequency spectrum and exposure limits vary accordingly [1]. Numerous techniques and methodologies have also been developed to assess the exposure of humans to EMFs [5,6,7], to demonstrate compliance with the aforementioned limits.

Radiofrequency EMF monitoring networks as a means of exposure assessment emerged about two decades ago. Numerous networks in several countries worldwide have been deployed [8,9,10,11]. Apart from being useful for demonstrating compliance and communicating results efficiently to the public, monitoring networks are a powerful tool for temporal analyses of EMFs both for short-term (e.g., day, week) [9] and long-term (years) variability [12]. In situ measurements can also be used in this context [13], while the large measurement campaign results available [14,15] include EMF exposure data for different microenvironments (e.g., indoor, outdoor, public places, and residences).

A comparative analysis of radiofrequency EMF exposure assessment has been previously performed at a European [16,17,18] and international [19] level, following a review scheme. In this paper, for the first time, the measurement data from the five largest monitoring networks currently operating in five different European countries (NOEF (Greece), SMRF (Catalonia in Spain), Observatoire des Ondes (France–Belgium), ANCOM (Romania) and RATEL (Serbia)), available online [10,15,20,21,22], were gathered and processed, along with the data from two extensive, ongoing in situ measurement campaigns [14,15], mainly in order to clarify some aspects resulting from the monitoring data. Variations in EMF over time and different microenvironments were studied, and the similarities/differences between different countries were identified. Correlations with demographic parameters, as suggested in [23], were also investigated to supplement this up-to-date study, which, for the first time, involves such an extensive dataset, granting its conclusions an added statistical value.

## 2. Materials and Methods

### 2.1. Data Acquisition

All the data used in this work are publicly available. One of the goals of RF-EMF monitoring networks and in situ measurements is to efficiently communicate measurement results to the public. For this reason, web portals have been developed [10,14,15,20,21,22], where citizens have access to the data and can be informed, using simple presentation tools (i.e., maps, graphs), of the EMF levels in their region or elsewhere.

Open data policies have been adopted by most operators of monitoring networks and in situ measurement campaigns considered here [14,24,25,26,27]. The data were accessed during February and March 2022, downloaded, and saved in a local MySQL (Oracle Corporation, Austin, TX, USA) database. In cases where open data are not available (e.g., Greece, Romania, Catalonia in Spain for data older than 6 months), access was either granted from the operator (Greece), or special software (using MATLAB (The MathWorks Inc., Natick, MA, USA)) was developed to download the data to the local database from the public web portal.

All the downloaded datasets, especially those acquired with software, were crosschecked with the published data, following an extended sampling procedure across different measuring sensors, installed in different regions, regarding different time periods of their lifetime.

### 2.2. Data Processing

The processing of the data was conducted with MATLAB and MySQL. MySQL offers an important speed advantage when processing large datasets, while MATLAB offers increased flexibility and a variety of tools to present the results obtained. Thus, the core of the software tools was developed in MATLAB environment, and appropriate MySQL queries were called and executed on the MySQL server-side whenever large data needed to be processed.

### 2.3. Data Description

#### 2.3.1. Continuous RF-EMF Monitoring Networks

A continuous RF-EMF monitoring network consists of several fixed measuring electric field (E-field) sensors, providing E-field measurement data on a 24/7 basis. Data, initially stored in local memory, are periodically (e.g., once a day) uploaded to a server, processed, and published through interactive web portals. Thus, data are made available to the public in a near-real-time manner. Measuring sensors are divided into 2 main categories: broadband and frequency-selective. Broadband sensors provide a single result for a wide frequency range (e.g., 100 kHz–7 GHz) where the vast majority of wireless communication systems operate. They are, sometimes, complementarily equipped with narrowband filters, capturing the E-field values of specific services (e.g., mobile telephony). Frequency-selective sensors, on the other hand, provide measurement results for several sub-bands (e.g., 20), usually reconfigurable, enabling more complex measurement campaigns and rendering the contribution of each wireless service to the total EMF level available. These advantages are offered at the cost of increased size and cost, compared with broadband sensors. As a result, the monitoring networks studied include a limited (if any) number of frequency-selective sensors. It is worth mentioning that the measurement results studied in this work exclusively concern broadband sensors and their broadband filter only. All the broadband sensors studied in this work have triaxial probes, thus performing isotropic measurements.

The largest monitoring network, to the best of the authors’ knowledge, is in Greece [10], named NOEF (National Observatory of Electromagnetic Fields). It consists of 480 broadband and 20 frequency-selective sensors. It is operated by the GAEC (Greek Atomic Energy Commission) and was deployed in 2015. Broadband sensors (AMB-8057, Narda STS, Pfullingen, Germany) measure the E-field values in the frequency range of 100 kHz–7 GHz. The sensitivity of the sensors is 0.2 V/m, The rms (root mean square) value of the E-field is logged every 6 min. (At the initial phase of deployment, the logging interval was 2 min.) Sensors are usually installed on the rooftops of buildings owned by local authorities (municipalities). The number of sensors installed in each municipality is proportional to its population. A few sensors have been relocated, so the data from a total size of 498 measurement sites were studied here.

SMRF (Sistema de Monitoratge de Radiofreqüència, i.e., Radio Frequency Monitoring System) was created in 2005 by the Catalan Government and was substantially upgraded in 2015 [15]. It now consists of 324 monitoring sensors installed at several municipalities in Catalonia, Spain. Monitoring sensors (MonitEM, Wavecontrol, Barcelona, Spain) have two different probe types: one measuring the E-field levels in 900 MHz, 1800 MHz, and 2100 MHz mobile telephony frequency bands with a sensitivity of 0.04 V/m and another measuring at the frequency range of 100 kHz–8 GHz with a sensitivity of 0.2 V/m. Most of the probes were of the first type initially, but in 2017, the Catalan Government decided to gradually substitute them with the second type. Thus, at the time the data were accessed (February 2022), only 37 sensors out of 324 were equipped with the probe measuring only at 900 MHz, 1800 MHz, and 2100 MHz frequency bands. This gradual transition, at a time instance for each sensor, unknown to the authors, imposes limitations when studying the temporal evolution of the E-field levels, as will be shown later. Sensors are usually installed on the rooftops of buildings as well.

In France–Belgium, the deployment of “Observatoire des Ondes” [22] started in 2020. In February 2022, it consisted of 161 monitoring sensors (custom-made, and EXEM, France). It is operated by the EXEM company (Toulouse, France) in cooperation with the National Frequency Agency of France (ANFR). The broadband monitoring networks measure the 6 min rms E-field value in the frequency range of 80 MHz-6 GHz. The logging interval is longer compared with other networks (2 h vs. 2 or 6 min), but the measurements are available online in real time. The measuring sensors are installed on street furniture (streetlamps), at a height of approximately 3–4 m from the ground level.

The RF-EMF monitoring network in Romania is operated by the National Authority for Administration and Regulation in Communications (ANCOM) [21]. The deployment of this network started in 2016. By 2019, 153 broadband measuring sensors (AMB 8057, Narda STS, Germany) were installed, which continuously measure the E-field in the frequency range of 100 kHz–7 GHz. The sensitivity of these sensors is 0.2 V/m. The logging time interval was 6 min at the initial phase and 2 min afterward. The sensors are mainly installed on the rooftops of buildings located in the vicinity of buildings of “sensitive usage” (i.e., hospitals, schools) or in the vicinity of EMF sources.

The deployment of the continuous RF-EMF monitoring network in Serbia started in 2017. It is operated by RATEL (Regulatory Agency for Electronic Communications and Postal Services). It now consists of 98 monitoring sensors (AMB 8057, Narda STS, Germany and MonitEM, Wavecontrol, Spain), mostly broadband, measuring the E-field value in the frequency range of 100 kHz–7 GHz. The sensitivity of the sensors is 0.2 V/m. The sensors are installed on the rooftops or facades of buildings of “increased sensitivity” (e.g., schools, hospitals, etc.) [20].

#### 2.3.2. In Situ Measurement Campaigns

In situ measurement campaigns involve spot EMF measurements at places of interest. Broadband or frequency-selective equipment is used to measure the E-field levels at a specific point in space and for a limited time interval (e.g., 6 min). The exact measurement point location is usually chosen as the one with the maximum E-field levels in the measurement site under investigation. Sometimes, E-field levels are re-evaluated over different time instances to capture temporal variations. This way, large measurement campaigns can be organized, addressing a variety of needs: compliance assessment near antenna sites, exposure assessment in different microenvironments (e.g., schools, hospitals, residences, indoors/outdoors, rural/urban, etc.), EMF exposure variations due to new communications technology rollouts (e.g., 5G), etc. In this work, the results of two extensive in situ measurement campaigns were studied [14,15].

In Catalonia, Spain, a large in situ measurement campaign took place between 2012 and 2019, while the vast majority of measurements were performed between 2013 and 2015. The E-field at 14,266 points located at 4979 different measurement places was evaluated. Fifty broadband radiation measurement systems have been distributed to different Catalan public bodies (mainly local councils), so they could take measurements in their municipalities, after appropriate training of their personnel [28]. The E-field levels in the frequency range of 100 kHz–8 GHz were measured in different microenvironments (e.g., kindergartens, pre-schools, primary and secondary schools, nursing and care homes, hospitals, private homes, etc.) both indoors and outdoors. The level of each measurement location is also available.

In France, a large dataset of in situ measurement campaigns is available online [14]. A total of 68,396 measurements were performed from 2001 until March 2022 during different measurement campaigns: (i) an ongoing campaign regarding citizens’ applications for an exposure assessment; (ii) campaigns organized by government and local authorities in outdoor locations (town hall squares), sensitive establishments (nursery and elementary schools, educational establishments, hospitals, retirement homes), or other public places (shopping centers, main train sensors) [29]; and (iii) a large measurement campaign following the 5G rollout. It is worth mentioning that only the broadband component of the measurements in the aforementioned campaigns was considered in this study.

## 3. Results and Discussion

The results of this study are organized into three different parts. Firstly, a comparative study of the results of the five continuous RF-EMF monitoring networks is presented. Then, the temporal evolution of the RF-EMF levels in Europe is evaluated, using the data from monitoring networks and in situ measurement campaigns. Finally, the results regarding different microenvironments, emanating from in situ measurement campaigns are also presented.

### 3.1. Comparative Study of Results from Five Monitoring Networks

At first, the E-field values measured from the measuring sensors of each country were processed. For each measuring sensor, the total rms (root mean square) E-field value in V/m (E_rms_), for the whole period of its installation, was evaluated. The rms value for time averaging was selected as the most appropriate averaging function to handle the different logging time intervals present in the data (even regarding the same station at different time periods). It is worth mentioning that E_rms_ is evaluated for different time periods in general, as the monitoring sensors were not installed concurrently. Nevertheless, the uncertainties induced by this fact are limited, since the temporal variations were not significant, as shown in Section 3.2.

A descriptive view of the distributions for each monitoring network is presented in the boxplot chart (Figure 1) and summarized in Table 1. None of the values exceeded the lowest ICNIRP (International Commission on Non-Ionizing Radiation Protection) reference level [1] for the general public (i.e., 27.7 V/m). However, differences between measurement results among different countries were obvious. A factor that could substantially differentiate results was the measurement location selection criteria, usually an unbalanced mixture of addressing the needs for compliance assessment and the exposure assessment of “sensitive places”. Lacking information on the exact criteria applied in each country, we used population density as an objective metric to determine the effect of sensor location. Other factors, such as per capita mobile subscriptions and the generation of the cellular network, are also expected to affect sensor recordings, depending on the location of the sensors. However, we did not find any literature source indicating large variations in mobile phone penetration rates within the countries studied here; therefore, we assumed that mobile phone user density is proportional to population density. Moreover, the indicators of the United Nations Economic Commission for Europe (UNECE) show that for all the countries studied in this work, population coverage with at least 4G technology reaches around 99% of the population [30]. The deployment of 5G is not expected to play a role in our analysis, in particular, since in these countries, the NSA (non-Stand Alone) architecture was used in the studied time period. The use of population density was chosen for the analysis as a good surrogate of urbanicity, which has been proven to play the most important role in exposure to environmental radiofrequency EMFs: in their systematic review, Jalilian et al. [16] concluded that “RF-EMF exposure, predominantly downlink signals, tends to increase with increasing urbanicity”.

In this context, the high-resolution population density data from the Humanitarian Data Exchange (HDX) were used [31]. Data contain population estimates for tiles of 30 m ×30 m, based on the building density of each tile (acquired from satellite images’ processing) and census data. These data were spatially averaged over a circular region with a radius of 250 m, surrounding each measuring sensor location, to obtain a more representative value of the population density of the area, as shown in Figure 2.

After the population density estimate of the area surrounding each measuring sensor was evaluated, the E-field measurements vs. the population density were compared between countries, the results of which are shown in Figure 3. E-field values increase as a function of increasing population density, in general. This trend has already been shown [9] but with a coarser characterization (i.e., rural/urban). Deviations from this trend observed in Figure 3 are attributed to the relatively small sample size of measuring sensors when individual networks were examined. The analysis of a much larger sample size (as will be shown later for the case of in situ measurements in France) confirmed this observed trend for fixed sensors.

In order to study the influence of the population density of the areas selected for the installation of measuring sensors among the different countries, we plotted the cumulative distribution functions (CDFs) for both the population density and the E-field (Figure 4).

Figure 4b shows that measuring sensors are installed in areas with different population densities among the different countries. The lowest population density is in Greece, followed by Romania, Serbia, Catalonia (Spain), and France–Belgium. Figure 4a shows that the E-field values measured follow roughly the same ranking, except for France–Belgium. This indicates that population density is an important factor that significantly affects E-field levels. In [32], urbanicity, an alternative for population density with coarser characterization (i.e., rural, suburban, and urban), was also identified as the most important determinant of both downlink and total exposure. However, the differences between countries can also include a contribution from the different power transmitted from base stations, which depends on the deployment strategy and network architecture in each country.

In the case of France–Belgium, the lowest E-field values were observed, while the population density was the highest. Another important factor that can affect measured E-field levels is the height of the position where the measuring sensor is installed: in France–Belgium, sensors are installed on streetlamps (at a height of 3–4 m from the ground level), while sensors in other monitoring networks are usually installed on the rooftops of buildings. This significant height difference is assumed to be a reason for this exception. The results presented in the case of in situ measurements in Catalonia in Spain (Section 3.3)) seem to confirm this assumption. Other measurement site selection criteria significantly affecting the measurement results could be (a) the sites located exclusively near emitting antennas vs. the sites selected randomly; (b) the sites located exclusively on buildings of specific usage (e.g., local authorities).

### 3.2. Temporal Evolution of RF-EMF Levels

In this section, the same dataset is used. The time averaging of the E-field is the same as in Section 3.1, but here, it is made on a yearly basis. The analysis was performed for each monitoring network separately since their operation period varies. The France–Belgium network was not included due to its short operation period.

For the distribution of E_rms_ (V/m), evaluated yearly for each sensor, the median (Q2), mean (i.e., rms), 25th (Q1), 75th (Q3), and 90th (Q90) percentiles were evaluated and plotted for each monitoring network, as shown in Figure 5. In order to distinguish the initial phase of deployment for each network, black verticals were drawn, and a grey-shaded background was added. Between the black verticals in the grey-shaded background, at least 75% of the measuring sensors were active for each monitoring network. The initial phase of deployment was not regarded as adequate for drawing conclusions about temporal evolution, since the uncertainties induced by the rapidly growing sample size were relatively large.

The results from the continuous monitoring networks in Greece (Figure 5a) did not show a significant trend in measured values, as can be inferred from the median value. An increase in the highest values (indicated by Q3 and Q90) from 2019 to 2021 could indeed be observed. This increase, which is also “responsible” for the corresponding increase in mean value, is in line with the trend observed from ANFR in situ measurements, presented in their annual report [33] (p. 11).

A moderate increase was initially observed in Catalonia (Figure 5b). All the values presented in the graph showed an increase in measured values from 2013 to 2021. However, this increase is probably caused by the fact that since 2017 until the present day, the probes of many sensors are gradually substituted from others of a broader frequency range [34]. Thus, in 2017, there were 281 probes in the frequency range of 900 MHz, 1800 MHz, and 2100 MHz, and 48 probes in the range of 100 kHz–8 GHz, while at the beginning of 2022, this was reversed, i.e., 37 sensors were equipped with probe types measuring only in the frequency bands of 900 MHz, 1800 MHz, and 2100 MHz, and 287 in the frequency range of 100 kHz–8 GHz. Since the “earlier probes” (900 MHz, 1800 MHz, and 2100 MHz) take measurements in a frequency range that is a subset of the frequency range of the “later probes” (100 kHz–8 GHz), they are expected to measure lower values. This is also confirmed by the reports [34,35,36,37,38,39], where the two probe types’ measurements are distinguished. In these reports, which, however, use different metrics, no significant trend was reported in 2013–2020. This confirms that the increase observed in Figure 5b is caused by the gradual substitution of the probes in the vast majority of measuring sensors, rather than by changes in the electromagnetic environment.

In Romania (Figure 5c), no significant trend was observed during the period 2019–2021. In Serbia (Figure 5d), the available interval was rather short (2020–2021). No significant variation for this interval can be observed in the graph.

The analysis of the RF-EMF levels’ yearly variation over the last few years did not show a significant trend toward any tendency. A mild increase in the higher values observed in the largest monitoring network (Greece) and a large in situ measurement dataset (France) [33] (p. 11) were identified.

### 3.3. EMF Levels in Different Microenvironments

The measurement results processed from the two large in situ measurement campaigns (Catalonia in Spain and France) are presented here. EMF levels were distinguished based on the microenvironment where they were measured.

In Catalonia, 14,266 measurements were performed at 4979 different places, mostly in the period 2013–2015. The places were categorized based on the type of microenvironment into nine different categories: (a) primary care centers, (b) nursing and care homes, (c) private homes, (d) compulsory secondary schools, (e) hospitals, (f) pre-school and primary schools, (g) kindergartens, (h) public parks, and (i) unclassified. The results are summarized in Table 2.

The values presented in Table 2 are regarding the measurements performed both indoors and outdoors. In Table 3, the measurements are categorized as outdoors–indoors. Table 3 refers to the measurements performed in any of the microenvironments referred to in Table 2.

None of the values measured exceeded the lowest ICNIRP reference level for the general public (i.e., 27.7 V/m) [1]. The measurements performed outdoors were of higher values, compared with those performed indoors. This is as expected, due to the attenuation from walls and other building materials, and is in line with the values measured in [40].

An increase in E-field as a function of height was observed in both indoor and outdoor environments (Figure 6). This is straightforward from the ground level (0 floors) to the 5th floor, for which the sample size was sufficient. The outdoor levels of E were higher than the indoor for each floor/level, as expected. It should be noted here that floor numbering alone can only give an indication of the height of measurement since different buildings (office, commercial, residential) can have different floor heights.

In France, 68,396 measurements of RF-EMF levels were studied, which were performed in the period from 2001 to March 2022 [41,42,43,44,45,46,47]. The measurements studied here were performed during several measurement campaigns in this period, and only the broadband component was taken into consideration, as already mentioned in Section 2.3.2.

A comparison of Table 4 and Table 5 to the corresponding ones in the case of Catalonia in Spain (Table 2 and Table 3, respectively) shows that the values measured in France were higher, in general. One possible reason could be the fact that the published values in France regarding a site are only the maximum ones measured in a few locations surrounding the site. On the other hand, in Catalonia, every measurement performed in the vicinity of a site is published and consequently processed.

Table 4 and Table 5 show that there are cases where the lowest ICNIRP reference level (i.e., 27.7 V/m) [1] was exceeded. In these cases, a frequency-selective measurement and extrapolation to maximum traffic was performed [41]. In those cases where extrapolated E-field levels exceeded the lowest ICNIRP reference level, appropriate actions (shutdown or power reduction in relative transmitters) were taken [33,46,47], measurements were repeated, and finally, compliance was demonstrated.

Finally, the correlations of the E-field levels with the population density of the area where the measurement was performed were studied (Figure 7). The methodology used is the same as that used in the case of monitoring networks, described in Section 3.1. The large sample size available here (68,396 measurements) enables a straightforward demonstration of this correlation and offers a good argument that the lower-than-expected E-field levels recorded by fixed sensors in France–Belgium can be attributed to the position (height) of these sensors.

The results of this work for the in situ measurements seem to be in good agreement with other studies. In [17], the range of RF-EMF identified for exposure in outdoor locations was (~0.3–0.7 V/m). The median values evaluated here for outdoor locations both for Catalonia (Spain) (0.3 V/m) and France (0.62 V/m) lay in the aforementioned interval. In [16], mean RF-EMF exposure levels in homes, schools, and offices were between 0.04 and 0.76 V/m. The median values in this work for these microenvironments were 0.33 V/m for homes and 0.2 V/m for schools (in the case of Catalonia in Spain) and 0.45 V/m for homes and 0.73 V/m for business places (in the case of France). The results of this work for the sensors of monitoring networks are generally higher (median value of 0.98 V/m for aggregated results over “Europe”, Table 1). This divergence can be attributed to the fact that the majority of these sensors are installed outdoors, at significantly higher altitudes (usually on the rooftops of buildings) than ground level, where most outdoor spot measurements are usually performed. Taking a closer look at Figure 6b for in situ measurements in Catalonia, for which the median E-field values vs. floor number was plotted, the median value on the 5th floor was 0.99 V/m, which seems to confirm the above assumption.

The strengths of this study are the extensive dataset collected and consequently processed and the fact that the same methodology and metrics were used for processing, unlike previous review studies, thus facilitating comparativeness. On the other hand, the lack of knowledge about the details concerning the dataset (such as the exact criteria applied for the selection of the sensors’ locations) and the differences existing among different monitoring networks and measurement campaigns impose certain limitations to this procedure, as does the lack of knowledge on the operators’ strategies in different countries when deploying and operating a cellular network.

## 4. Conclusions

The continuous monitoring of RF-EMF networks and in situ measurement campaigns are tools largely used for the demonstration of the compliance of environmental EMF with safety limits for the general public. In Europe, five large monitoring networks (Greece, Catalonia in Spain, Romania, Serbia, and France–Belgium) are currently operating, consisting of more than 1200 measuring sensors. Continuous data from monitoring networks are very useful for the analysis of the temporal evolution of exposure [9,12]. Two large in situ measurement campaigns (Catalonia in Spain and France) also provide useful information on exposure. In situ measurement campaigns provide useful data for the variation in exposure in different microenvironments where people spend most of their time [16].

The population density of an area is a factor correlated with E-field levels: The higher the population density, the higher the E-field measured. This can be concluded from the measurements of monitoring networks, where we observed higher E-field values in those countries where the sensors are located in places with higher population density. This conclusion was confirmed from in situ measurements within the same country (France), where the correlation was straightforward due to the large sample available.

However, the population density was not the only factor affecting measurement results. As shown in the case of the monitoring network in France–Belgium, sensor positioning was also an important factor: The higher the sensor was placed, the higher the E-field levels measured. This conclusion was confirmed by the in situ measurement campaign results in Catalonia in Spain, where the level/floor of each measurement location is available.

Differences exist in the design and deployment among different monitoring networks and in situ measurement campaigns. These differences can impose several problems and be misleading when trying to draw general conclusions. Such problems, among others, were faced in the current analysis of results in the cases of (i) the France–Belgium monitoring network (due to different sensor positioning compared with other networks) and (ii) the SMRF monitoring network (Catalonia, Spain) due to the gradual substitution of the probes with others operating in a broader frequency range. Indeed, proper reporting of relevant parameters can mitigate these problems. As an example, we can mention the reporting of (i) indoor/outdoor available information in France’s in situ measurement campaign that enabled the demonstration of E-field vs. population density correlation and (ii) the extent of the information available for measurement location (i.e. level) in the case of Catalonia’s in situ measurements that enabled the demonstration of E-field vs. level/height correlation. In this context, a common framework for the development and operation of monitoring networks and in situ measurement campaigns could homogenize measurement results among different countries and significantly facilitate the attainment of useful conclusions.

## Figures and Tables

**Figure 1 sensors-22-08481-f001:**
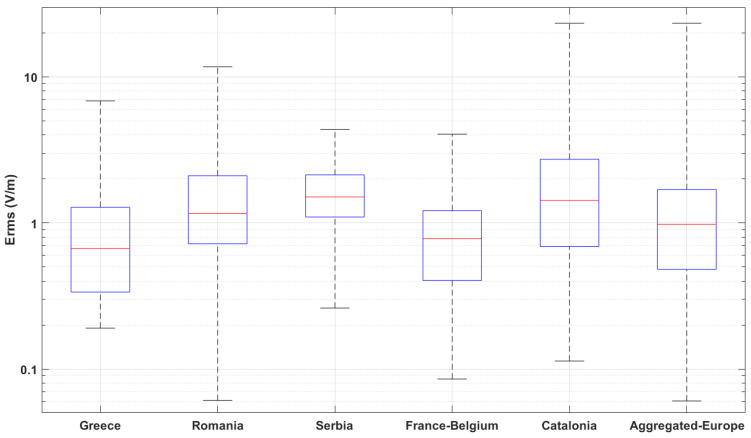
Boxplot charts for the distributions of monitoring sensors’ Erms (V/m) value: median values (red lines), interquartile ranges (blue boxes), and full range (min–max) of distributions (dashed lines).

**Figure 2 sensors-22-08481-f002:**
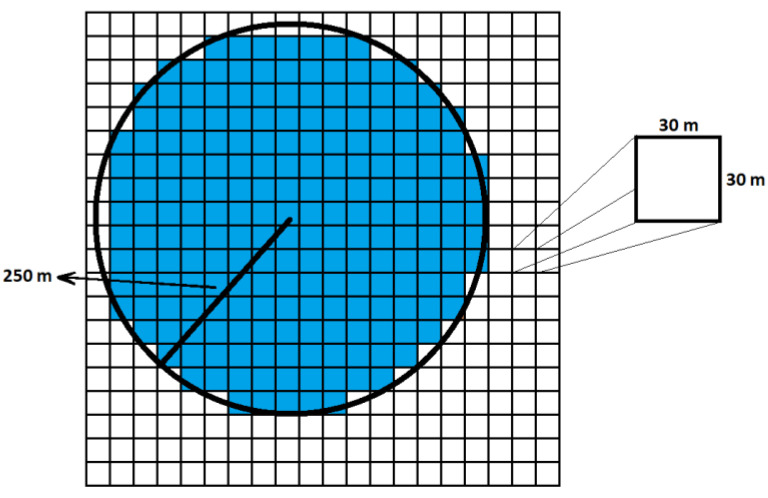
Population density estimation of the circular area (radius = 250 m) surrounding each measuring station’s location (center of circle). High-resolution (per 30 m × 30 m tiles) population density data from HDX were used and averaged over the circular area (blue-shaded tiles).

**Figure 3 sensors-22-08481-f003:**
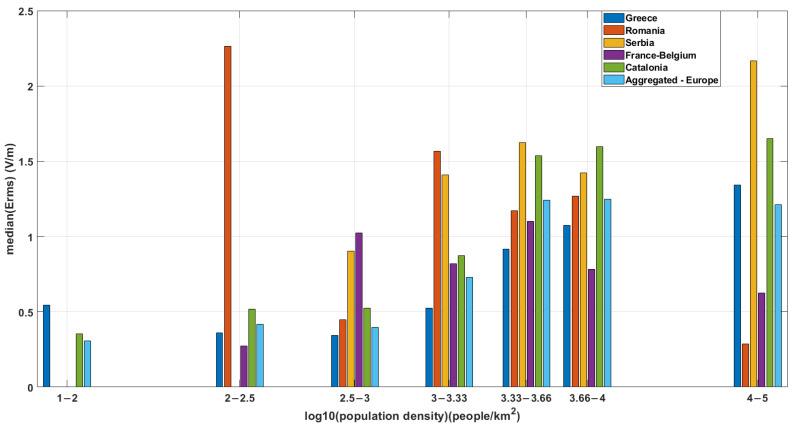
Median E_rms_ (V/m) vs. log_10_ (population density) grouped by country. Aggregated results are denoted as “Europe”. The figure shows a trend of increasing median electric field with increasing population density.

**Figure 4 sensors-22-08481-f004:**
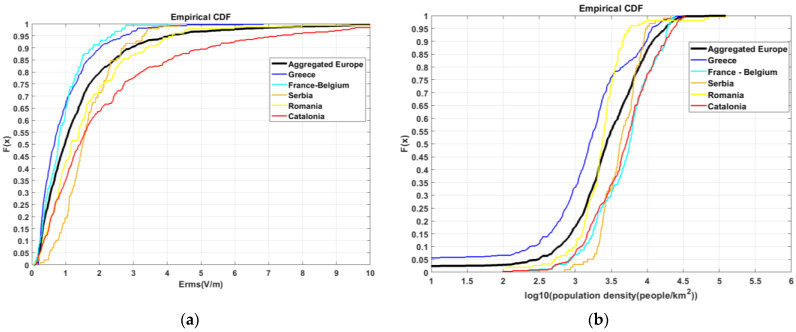
Cumulative density functions (CDFs) for (**a**) distribution of measuring sensors’ E_rms_ values and (**b**) distribution of measuring sensors’ surrounding area population density.

**Figure 5 sensors-22-08481-f005:**
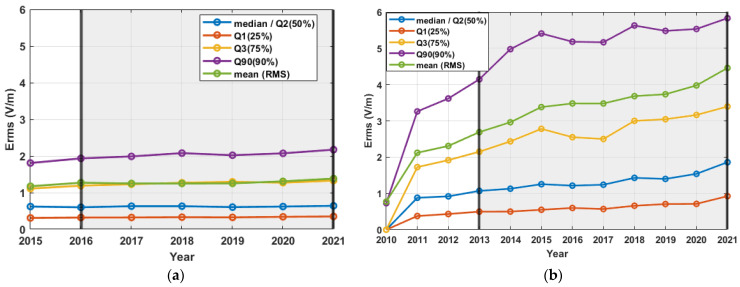
Temporal evolution of E-field levels for four monitoring networks: (**a**) Greece, (**b**) Catalonia in Spain, (**c**) Romania, and (**d**) Serbia. Median, mean (rms), and several percentiles’ values for the yearly E_rms_ distribution of each network are shown. The time period where at least 75% of monitoring sensors are active is indicated between black vertical lines in grey-shaded background, for each network.

**Figure 6 sensors-22-08481-f006:**
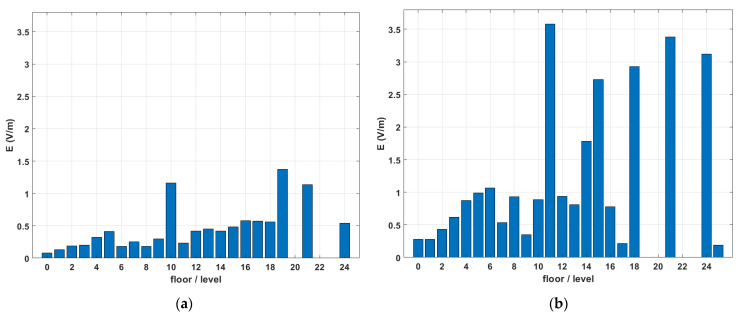
Median value of E (V/m) values measured in various microenvironments in Catalonia in Spain: (**a**) indoors; (**b**) outdoors.

**Figure 7 sensors-22-08481-f007:**
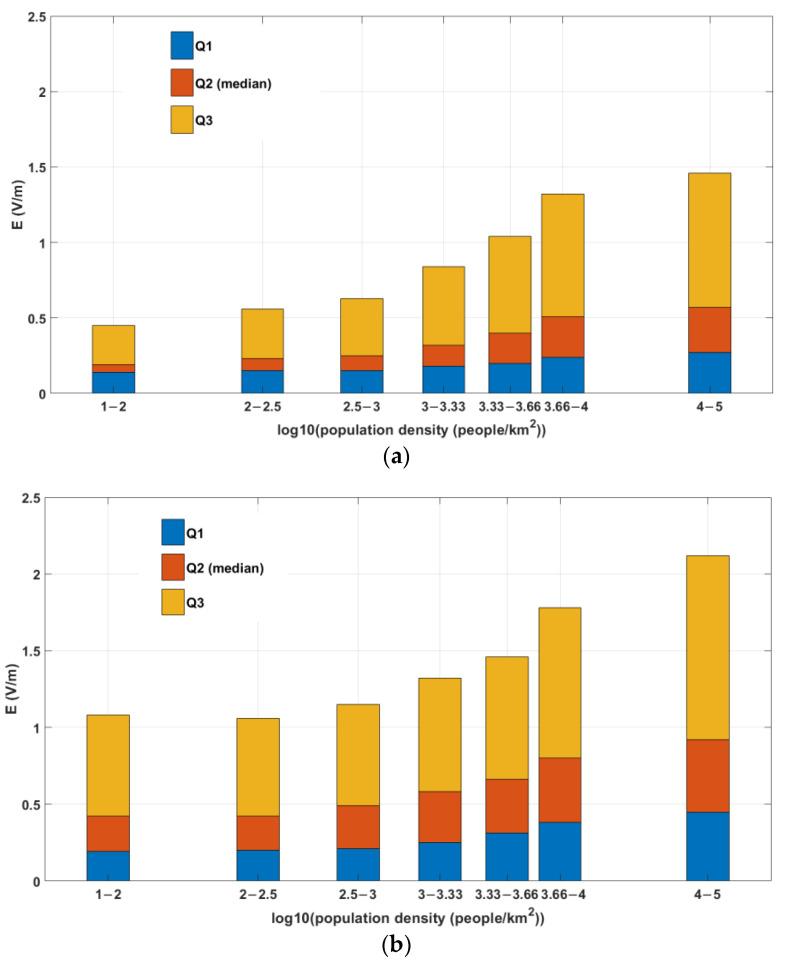
Median value, 25th, and 75th percentiles of E (V/m) values measured in France vs. log_10_ (population density): (**a**) indoors; (**b**) outdoors.

**Table 1 sensors-22-08481-t001:** Descriptive statistics parameters (start of operation, 75% sensor activation period, median, 25th and 75th percentiles, minimum and maximum) for the E_rms_ (V/m) values of monitoring sensors grouped by country.

Country	Start of Network Operation	Period of 75% Sensor Activation	Median	Q1 (25%)	Q3 (75%)	Min	Max
Greece	2015	2016–2022	0.67	0.34	1.28	0.19	6.85
Romania	2016	2019–2022	1.16	0.72	2.11	0.06	11.74
Serbia	2017	2020–2022	1.51	1.10	2.13	0.26	4.36
France–Belgium	2020	2021–2022	0.78	0.40	1.22	0.09	4.06
Spain (Catalonia)	2005	2013–2022	1.43	0.69	2.73	0.11	23.32
Aggregated-Europe	-	-	0.98	0.48	1.70	0.06	23.32

**Table 2 sensors-22-08481-t002:** Descriptive statistics parameters (median, 25th, and 75th percentiles and maximum) for the E (V/m) values measured in various microenvironments in Catalonia (Spain).

Microenvironment	Median	Q1 (25%)	Q3 (75%)	Max
Primary care centers	0.10	0.01	0.25	3.21
Nursing and care homes	0.14	0.03	0.35	6.21
Private homes	0.33	0.13	0.83	16.54
Compulsory secondary schools	0.21	0.06	0.41	4.84
Hospitals	0.28	0.05	0.68	8.20
Pre-school and primary schools	0.20	0.04	0.40	5.91
Kindergartens	0.16	0.03	0.29	3.75
Public parks	0.31	0.09	0.64	3.40
Unclassified	0.28	0.10	0.59	20.01

**Table 3 sensors-22-08481-t003:** Descriptive statistics parameters (median, 25th, and 75th percentiles and maximum) for the E (V/m) values measured in indoor and outdoor places in Catalonia (Spain).

Place	Median	Q1 (25%)	Q3 (75%)	Max
Indoors	0.14	0.03	0.31	13.09
Outdoors	0.30	0.12	0.63	20.01

**Table 4 sensors-22-08481-t004:** Descriptive statistics parameters (median, 25th, and 75th percentiles and maximum) for the E (V/m) values measured in various microenvironments in France.

Microenvironment	Median	Q1 (25%)	Q3 (75%)	Max
Business (shopping centers, etc.)	1.42	0.46	2.95	16.40
Various	0.70	0.30	1.47	31.05
Establishments open to the public (schools, care facilities, sensors, etc.)	0.36	0.20	0.85	28.30
Residences	0.45	0.21	0.95	40.86
Business places not accessible to the public	0.73	0.35	1.81	19.59
Streets/roads/parking lots	0.61	0.28	1.10	31.56

**Table 5 sensors-22-08481-t005:** Descriptive statistics parameters (median, 25th, and 75th percentiles, and maximum) for the E (V/m) values measured in indoor and outdoor places in France.

Place	Median	Q1 (25%)	Q3 (75%)	Max
Indoors	0.41	0.20	0.93	40.86
Outdoors	0.62	0.28	1.16	31.56

## Data Availability

Publicly available datasets were analyzed in this study. This data can be found here: https://www.cartoradio.fr, https://politiquesdigitals.gencat.cat/ca/tic/governanca/, https://paratiritirioemf.eeae.gr/en/, https://emf.ratel.rs/, https://www.monitor-emf.ro/map, https://www.observatoiredesondes.com/fr/, (accessed on 25 September 2022).

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
