# Peer review of "Electromagnetic Fields Exposure Assessment in Europe Utilizing Publicly Available Data"

_sensors, 2022, doi:10.3390/s22218481_

Round 1

Reviewer 1 Report

The article presents the review and statistics of the electromagnetic field exposure in different locations in the Europe. The article in general is interesting and relevant however I have two doubts to the article:

From line 206 to 254: This part is not clear to me. I understand that Authors tried to find (probably) the relation between population density and EMF, however in my opinion, it does not work as simple as this. The EMF depends of course partially on simple population density, however there are more aspects such as: development of the society living there (more mobile phones and electrical equipments), development of the area (more offices vs. more factories), development of applied technology (some cities have already 5G while some of them does not have even 4G) and development of the IT infrastructure (some cities have internet via 4G - 5G while some of them have very fast optofiber internet without EMF emissivity).  

Moreover, we do not know the exact location of the sensors. Some of them could be put only in the cities while some of them in cities and villages which can strongly undermine Authors statement that EMF depends on simple population density. It is just one of many factors. 

In my opinion, the research should be massively improved from lines 206 to 254. If it is impossible, I suggest removing this controversial part from the manuscript. 

The second doubt is that the introduction does not contain information about other research on the EMF exposure in the past. The EMF exposure has been investigated many times e.g. in  10.3390/ijerph18041730, where Authors tried to find relation between EMF exposure and particular occupation. Here: 10.1016/j.envres.2019.05.048 is another example of the EMF exposure study across the Europe. You should provide complex state of the art regarding to the current reviews and explain what is new in your article. 

Some detailed comments:

Lines 28-30: I recommend adding some examples of the health risks with the reference literature. 

Line 34: I suggest providing here the information that different countries in Europe have different limits of EMF field and also providing some examples (with reference). 

Please mention also in the introduction, that the health risks depend not only on the amplitude of EMF and time of exposure but also on the frequency of the EMF. 

You use "Catalonia" in comparison with "countries". I think it can be politically risky due to Catalonia is not independent on Spain. I suggest  exchanging "Catalonia" into "Spain" or "Catalonia in Spain". 

Line 83 and the rest of the article: Please put a space between number and unit. 

Line 186: The ICNIRP acronym was not explained. Please explain here also how the level of ICNIRP is defined. 

Figure 1 and 2: In my opinion the Figure 1 can be removed. The distribution of the measurement results in the figure 1 is of course Gaussian. All necessary data is provided by the Figure 2. 

From line 190: "Lacking information on the exact criteria applied in each country, we used population density as an objective metric to determine the effect of sensor location." - As I understand correctly, the Authors mean: the higher population density = the higher density of EMF sources. Please clarify this in the text. 

Figure 3: I do not understand this figure. 

Figure 7: It is better to use log y scale rather than two plots. 

Table 2 and Figure 7: In my opinion Table 2 and Figure 7 presents the same. Authors should decide and choose one of them.

Fig. 9 and Table 4: similar to above.

Reviewer 2 Report

GENERAL COMMENTS

 The work is interesting, because it presents data from the five largest monitoring networks currently operating in five different European countries, however, in order to be published the authors must make major changes.

  SPECIFIC COMENTS

 The authors could calculate the intensity that corresponds to the electric field shown in the results, review and cite the following article: Ramirez-Vazquez, R., Escobar, I., Franco, T., Arribas, E., 2022. Physical units to report intensity of electromagnetic wave. Environ. Res. 204, 112341. https://doi.org/10.1016/j.envres.2021.112341

 In the Methodology section, you should specify which measurement equipment is used by each of the monitoring networks.

 Line 43, indicate the name of the five largest monitoring networks currently operating in five different European countries.

 Line 203, Fig. 1, to which period does it refer, it is necessary to specify in each graph the period in which the measurements have been taken. It would be interesting to show in a separate table the descriptive results of the measurements for each of the monitoring networks.

 Fig. 7 and 9 are not very clear, they should be corrected, the x-axis is not visible.

 Explain in more detail what Fig. 3 represents and why Fig. 4 is included.

 Some graphs need to show data labels, for example in Figure 4 I can't see the exact value of the median of E.

 A discussion section is needed in which the advantages and disadvantages of the study are indicated, and these results are compared with other Radio Frequency Electromagnetic Field measurement studies.

 With the large amount of measurement data obtained by the authors, I believe that this article could be more complete.

Reviewer 3 Report

This paper introduces electromagnetic field exposure assessments in Europe, based on sensor monitoring networks and in situ measurements.

For the sensor monitoring networks in different EU countries, an important fact is the activation time of the day for each sensor. Since this information can help people to understand better the uncertainty in the sensor measurements, e.g, some sensor may perform measurement once per day, some may perform several times regularly in a day.

In terms of the sensor network design, are the antennas of sensors isotopical? or 3 axis to mimic omni-directional reception?

In table 1, the statistic parameters of monitoring sensors are shown. It would be better to add the years when the sensors are in service, such as, 20XX to 20XX. Since it may include different generations of technology. Also, the noise level should be mentioned in the text, which will provide a reference level to understand minimum value in the table.

The significant part in Figure 1 is too small, maybe a histogram with showing only up to 95% or 99% of all Erms values will give clearer distributions. Also the max values are already given by Table 1, so that there is some overlap.

Figure 2 and 3 are too big, compared with Figure 1. In captions of Figure 2, it is mentioned dashed lines represent range of distributions, please define it. Since sometimes dashed lines in a boxplot are estimated by ‘Minimum’ Q1-1.5IQR and ‘Maximum‘Q3+1.5IQR, where IQR=Q3(75th percentile)-Q1(25th percentile), which means not true min and max of the data.

Figure 4 and 5, another factor may influence the distribution is that, the transmit power of each base station antenna may be different from country to country.

In Figure 6, the 75% active sensor region is not clear by adding only the black line, it will be good to denote the whole region by shade or gray color. In sub-Figure 6 (b), the increasing trend comes from the replacement of broadband sensors, then why not only put broadband sensor results here? Since this comparison (3 almost constant, 1 increasing) from different countries will cause confusion.

Figure 7 and Table 2 contain same information, maybe keeping one of them is enough. Same with Figure 9 and Table 4. Xlabel fontsize in Figure 7 and 9 is too small to recognize.

In Figure 8, what does floor/level mean? They are measured in the same building structure? residential or office building? Since different buildings may have different floor heights.

Round 2

Reviewer 1 Report

The manuscript has been revised. The letter to the reviewer explained doubts. Now the manuscript can be published in the present form. 

Reviewer 2 Report

It appears that you have addressed and considered most of the issues identified by me  and the paper has been improved.  Therefore, I am recommending the acceptance of your manuscript.